# Towards Scalable Bayesian Learning of Causal DAGs

**Jussi Viinikka**
Department of Computer Science
University of Helsinki
jussi.viinikka@helsinki.fi

**Antti Hyttinen**
HIIT & Department of Computer Science
University of Helsinki
antti.hyttinen@helsinki.fi

**Johan Pensar**
Department of Mathematics
University of Oslo
johanpen@math.uio.no

**Mikko Koivisto**
Department of Computer Science
University of Helsinki
mikko.koivisto@helsinki.fi

## Abstract

We give methods for Bayesian inference of directed acyclic graphs, DAGs, and the induced causal effects from passively observed complete data. Our methods build on a recent Markov chain Monte Carlo scheme for learning Bayesian networks, which enables efficient approximate sampling from the graph posterior, provided that each node is assigned a small number $K$ of candidate parents. We present algorithmic techniques to significantly reduce the space and time requirements, which make the use of substantially larger values of $K$ feasible. Furthermore, we investigate the problem of selecting the candidate parents per node so as to maximize the covered posterior mass. Finally, we combine our sampling method with a novel Bayesian approach for estimating causal effects in linear Gaussian DAG models. Numerical experiments demonstrate the performance of our methods in detecting ancestor–descendant relations, and in causal effect estimation our Bayesian method is shown to outperform previous approaches.

## 1 Introduction

Bayesian learning of graphical models aims at assigning any event of interest a posterior probability given observed data over the variables. In causal directed acyclic graph (DAG) models, examples of such events include presence of a causal path between two variables and the total causal effect of one variable on another. While the posterior of the former event is quantified by a single number, the latter is represented by a distribution. The Bayesian approach is particularly attractive in the causal setting due to its ability to properly account for the often non-negligible uncertainty in the inferred causal structure. In comparison, non-Bayesian structure learning methods are more limited in this aspect, as they typically return a single DAG, or Markov equivalence class, without any associated measure of uncertainty. In the case of linear Gaussian models, the prospects of the Bayesian approach have recently been demonstrated [31, 2], showing an improved estimation accuracy over the original non-Bayesian IDA method [22] and some of its later variants. However, the power of Bayesian learning stems from model averaging which unfortunately has appeared to be computationally intractable in the combinatorial space of DAGs. Hence, the currently existing and provably accurate algorithms are feasible only with up to around 25 variables [14, 39, 37, 31], and algorithms with somewhat looser accuracy guarantees to several dozens of variables [20].

There have been several attempts to scale up Bayesian learning of graphical models using Markov chain Monte Carlo (MCMC). The first methods simulated a Markov chain on the space of DAGs by applying edge operations (add, remove, and reverse edge), yielding a sample of DAGs approximately from the posterior [24, 11]. To improve the sampler's ability to escape from local optima, subsequent

Table 1: Space and time requirements with $n$ nodes and $K$ candidate parents per node

| Task | Space | Time | Previous work [19] |
|---|---|---|---|
| Pre-processing | $O(3^K + 2^K n)$ | $O(3^K n)$ | $O(3^K n)$ space, $O(3^K K^2 n)$ time |
| Simulation step | $O(2^K n)$ | $O(n)$ | $O(3^K n)$ space |
| Sampling $r$ DAGs | $O(3^K + Knr)$ | $O(3^K n + Knr)$ | $O(2^K nr)$ time |

works collapsed the space of DAGs to linear and partial node orderings covering multiple DAGs [7, 29]. While the smaller state space and smoother posterior landscape enhanced the reliability of the order-based samplers, they still suffered from two major drawbacks. First, the sampling distribution is *biased*, favoring graphs that are compatible with a larger number of orderings. This is particularly problematic in the causal setting, since the bias forces one to assign a nonuniform prior over equivalent DAGs. Markov chains directly on equivalence classes suffer, again, from the large, combinatorial state space [23, 3]. Second, each simulation step is *computationally expensive*, since it requires summing over the local scores of all order-compatible parent sets for each node. This issue is emphasized in linear Gaussian models, where also larger parent sets are more probable a priori, as the number of parameters for a node grows only linearly with the number of parents.

The two issues were partly resolved in two recent works [16, 19]. The sampling bias was avoided by sampling ordered node partitions instead of node orderings. The per-step computational cost, in turn, was dramatically reduced by restricting the parents to a small candidate set (a technique also proposed earlier [7]) and, importantly, precomputing all possible score sums and storing them in a lookup table. Inspired by this progress, we here make several contributions to further advance the machinery and its applicability to causal inference. Specifically, we address the following questions.

Q1 *How many candidate parents can we afford?* The number of candidate parents per node, $K$, is a critical parameter. We wish to let $K$ be as large as possible to cover well the space of DAGs; unfortunately, the memory requirements and preprocessing time grow exponentially in $K$. We present several algorithmic ideas to reduce the space and time requirements, and thereby, to allow for a substantially larger $K$; see Table 1. Put otherwise, for fixed, practical values of $K$ and the number of nodes $n$, the savings are by 2–3 orders of magnitude compared to previous work.

Q2 *How to select the candidate parents?* The method assumes that we can select a moderate number of candidate parents per node, say $K = 15$, such that the posterior mass of DAGs is concentrated on the restricted family of DAGs, even if the number of nodes $n$ is much larger than $K$. We study to what extent this assumption holds by (i) formulating the selection task as an optimization problem, (ii) giving an exact algorithm to solve the problem optimally for moderate $n$, and (iii) introducing and empirically comparing various scalable heuristic algorithms to find good solutions when $n$ is large.

In addition to the above contributions and building upon our sampling method, we introduce a novel Bayesian approach for estimating causal effects in linear Gaussian DAG models with unknown causal structure, a subject of recent intensive ongoing research [22, 21, 36, 38, 31, 2].

Q3 *How to obtain the posterior of causal effects?* In a Bayesian linear DAG model, the posterior of a causal effect is obtained by integrating over the unknowns (structure and parameters). We propose a three-stage sampling-based method to approximate the posterior: (i) we sample a DAG using our proposed sampling method, (ii) we sample the model parameters conditional on the DAG, and (iii) we map the model parameters to their implied causal effects using a matrix inversion technique. Importantly, the key novelty in our estimator compared to the IDA approach is to make use of the complete DAG structure in the estimation procedure. Figure 1 shows example posterior distributions obtained by this method.

Like previous works [22], we assume the data to be complete in the sense that there are no hidden variables (faithfulness and causal sufficiency). The scalability of our methods allows us to present the first empirical comparison of the Bayesian approach to non-Bayesian methods in higher dimensions.

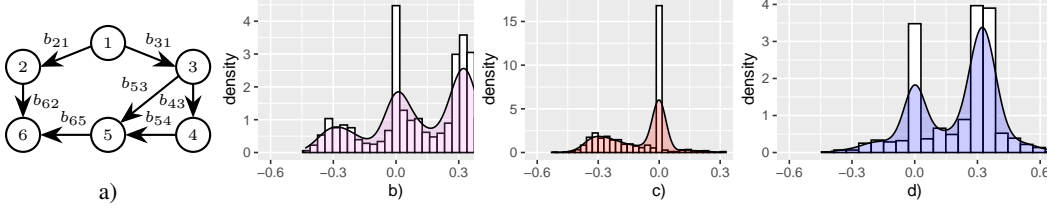

Figure 1: (a) A linear DAG model (error variances not shown). (b–d) The posteriors of the linear causal effect of $x_1$ on $x_6$ given observational data, when intervening on $\{x_1\}$ in (b), $\{x_1, x_2\}$ in (c), and $\{x_1, x_3\}$ in (d). The posterior in (b) is a mixture of the posteriors in (c) and (d).

## 2 Preliminaries

We shall use the following notational conventions. For a tuple $(t_1, t_2, \ldots, t_k)$ we may write shorter $t_1 t_2 \cdots t_k$ or $(t_i)$, or just $t$. If $S$ is a set, we write $t_S$ for the tuple $(t_i : i \in S)$.

A *directed acyclic graph (DAG)* $(V, E)$ consists of a node set $V$ and an edge set $E \subseteq V \times V$ that contains no directed cycles. If $ij \in E$, call $i$ a *parent* of $j$ and, conversely, $j$ a *child* of $i$. Denote the set of parents of $j$ by $\mathrm{pa}_G(j)$, or by $pa(j)$ when understood as a variable through the referred DAG. If there is a directed path from $i$ to $j$, call $i$ an *ancestor* of $j$ and, conversely, $j$ a *descendant* of $i$.

For a vector of random variables $\boldsymbol{x} = x_1 x_2 \cdots x_n$, a *Bayesian network (BN)* is a pair $(G, f)$, where $G$ is a DAG on the index set $V = \{1, 2, \ldots, n\}$ and $f$ is a joint distribution that factorizes along $G$ as $f(\boldsymbol{x}) = \prod_{i=1}^{n} f(x_i | x_{pa(i)})$. Specific representations of the conditional distributions yield more concrete models [15]. Among the most popular models are *discrete BNs*, in which the support of each variable is finite with fully parameterized conditional probabilities, and *linear Gaussian DAGs* [41, 8], in which the local distributions are Gaussians. The latter corresponds to a structural equation model $\boldsymbol{x} := \boldsymbol{\mu} + B(\boldsymbol{x} - \boldsymbol{\mu}) + \boldsymbol{e}$, with $\boldsymbol{e} \sim \mathcal{N}(\boldsymbol{0}, Q)$. Here $Q$ is a diagonal matrix of the error term precisions and $B = (b_{ij})$ a matrix of edge weights. The joint distribution of $\boldsymbol{x}$ is then $\mathcal{N}(\boldsymbol{\mu}, W)$, with the precision matrix $W = (I - B)^\mathsf{T} Q (I - B)$.

When a BN $(G, f)$ is interpreted as a causal model, $G$ encodes a hypothesis of the direction of causal relations. From $G$ alone, we can read off whether a node $j$ is an ancestor of $i$, and thus $x_j$ potentially has a causal effect on $x_i$. The magnitude is specified by the distribution $f$. We will focus on linear Gaussian DAGs, in which the *causal effect* of $x_j$ on $x_i$ is quantified by a single scalar $a_{ij}$ obtained by summing up the weights of all directed paths from $j$ to $i$, the weight of a path equalling the product of the coefficients associated with the edges. In Figure 1(a), node 1 is an ancestor of node 6 and $a_{61} = b_{62} b_{21} + b_{65} (b_{53} + b_{54} b_{43}) b_{31}$.

To learn a BN $(G, f)$, we assume $N$ independent samples $\boldsymbol{x}_1, \boldsymbol{x}_2, \ldots, \boldsymbol{x}_N$ from $f$. We denote by $X$ the $N \times n$ data matrix. We take a Bayesian approach and specify a joint distribution $p(G, f, X)$ as the product of the priors $p(G)$ and $p(f|G)$ and the likelihood $p(X|G, f) = \prod_s f(\boldsymbol{x}_s)$. We assume the priors satisfy standard modularity properties, so that the posterior of $G$ can be written as

$$p(G|X) \propto \pi(G) := \prod_{i=1}^{n} \pi_i\big(\mathrm{pa}_G(i)\big), \quad \text{with} \quad \pi_i(S) := \rho_i(S)\, \ell_i(S), \tag{1}$$

where $\rho_i$ and $\ell_i$ are factors of the DAG prior and the marginal likelihood: $p(G) \propto \prod_i \rho_i\big(pa(i)\big)$ and $p(X|G) = \prod_i \ell_i\big(pa(i)\big)$. For example, in our experiments we put $\rho_i(S) = 1/\binom{n-1}{|S|}$ and composed the prior $p(f|G)$ from conjugate priors so that $\ell_i(S)$ admits a closed-form expression that is efficiently evaluated for any given node set $S$, and that yield the marginal likelihoods known as the *BDe* and *BGe scores* for discrete and Gaussian models, respectively. With these choices the posterior $p(G|X)$ is *score equivalent*, meaning that the posterior probability is the same for Markov equivalent DAGs.

## 3 Scalable sampling of directed acyclic graphs

To draw a sample of DAGs approximately from the posterior distribution, we adopt the approach of Kuipers et al. [16, 19], implemented in the *BiDAG* package, with some major modifications.

**Algorithm 1** The *Gadget* method for sampling DAGs
___
1: *Preprocessing.* Select a set of candidate parents $C_i$ for each node $i \in V$. Build a data structure that enables fast evaluation of $\tau_i(U, T)$ for any $i \in V, T \subseteq U \subseteq V \setminus \{i\}$.
2: *Markov chain simulation.* Generate a realization of a Markov chain $R^0, R^1, \ldots, R^L$ whose stationary distribution is the posterior of root-partitions on $V$ using the Metropolis–Hastings algorithm. Store every $n$th sample $R^s$.
3: *Postprocessing.* Generate a DAG $G^s$ per sampled and stored $R^s$.
___

### 3.1 Outline

The basic idea is to sample DAGs by simulating a Markov chain whose stationary distribution is the posterior distribution. However, to enhance the mixing of the chain, we build a Markov chain on the smaller space of ordered partitions of the node set, each state being associated with multiple DAGs.

Let $R = R_1 R_2 \cdots R_k$ be an ordered set partition of $V$. We call $R$ the *root-partition* of a DAG $G$ if $R_1$ consists of the root nodes of $G$, $R_2$ consists of the root nodes of the residual graph $G - R_1$, and so forth; here $G - R_1$ is the graph obtained by removing from $G$ the nodes in $R_1$ and all incident arcs. Note that a DAG has a unique root-partition, whereas there may be several topological orders. For example, the root partition of the example DAG in Figure 1(a) is $\{1\}\{2,3\}\{4\}\{5\}\{6\}$.

The root-partition of $G$ is $R$ exactly when every node in $R_1$ has zero parents and every node in $R_t$, with $t \geq 2$, has at least one parent from the previous part $R_{t-1}$ and the rest from the union $R_{1,t-1} := R_1 \cup R_2 \cup \cdots \cup R_{t-1}$. This is also evident in Figure 1(a). Thus, by the factorization (1), the posterior probability of $R$, i.e., the total probability of DAGs with root-partition $R$, is given by

$$\pi(R) = \prod_{t=1}^{k} \prod_{i \in R_t} \tau_i(R_{1,t-1}, R_{t-1}), \quad \text{with} \quad \tau_i(U, T) := \sum_{S \subseteq U : S \cap T \neq \emptyset} \pi_i(S).$$

In words, $\tau_i(U, T)$ is the sum of local scores of node $i$ over all parents sets that contain at least one parent from $T$ and the rest from $U$. The factorization enables fast evaluation of $\pi(R)$, provided that the score sums $\tau_i(R_{1,t-1}, R_{t-1})$ can be computed fast. A fast evaluation is crucial for the scalability of the method, as the evaluation is required in every simulation step of the Markov chain.

The key observation is the following [19]. If node $i$ can only take parents from a small candidate parent set $C_i$, then it is feasible to precompute the needed values $\tau_i(U, T)$, for they only depend on the intersections $U \cap C_i$ and $T \cap C_i$. The evaluation then corresponds to a (nearly) constant-time table lookup. In Figure 1(a), we might discover that $C_1 = \{2, 3\}$, $C_2 = \{1\}$, $C_3 = \{1, 4, 5\}$, $C_4 = \{3, 5\}$, $C_5 = \{3, 4\}$, and $C_6 = \{2, 5\}$ are good choices for the candidate parents by simple linear regression.

Finally, we generate DAGs conditionally on the sampled partitions. Generating a single DAG by enumerating all possible parent sets would require time $O(2^K n)$ [19], which is expensive. Instead, we will generate DAGs as postprocessing in time $O(Kn)$ per DAG, by investing $O(3^K)$ space.

Algorithm 1 outlines the three phases of our method, we dub *Gadget* (Generating Acyclic DiGraphs Efficiently from Target). We describe the phases in more detail the remainder of this section.[1]

### 3.2 Preprocessing

In what follows, we assume that each node $i$ is assigned a set of candidate parents $C_i$ of size $K$. We will consider the task of selecting the candidate parents for each node in Section 4.

We aim at building a data structure that enables fast evaluation of the node-wise score sum $\tau_i(U, T)$ for any given $i, U, T$. To this end, for any $i \in V$ and $J \subseteq V \setminus \{i\}$, let

$$\tau_i(J) := \sum_{S \subseteq J \cap C_i} \pi_i(S),$$

the sum of all local scores for node $i$ with parents from $J \cap C_i$. Clearly, $\tau_i(J) = \tau_i(J \cap C_i)$. Furthermore, the values $\tau_i(J)$ are sufficient for instant evaluation of a score sum, by subtraction:

___
[1] For the sake of exposition, we here consider simplifications of *BiDAG* and *Gadget* that require all parents be from the $K$ candidates. In experiments we ran extended versions: *BiDAG* additionally allows one parent outside the candidates, and *Gadget* any three or fewer parents; using known techniques [7, 28] this is still feasible.

**Lemma 1.** *Let $i \in V$ and $T \subset U \subseteq V \setminus \{i\}$. Then $\tau_i(U, T) = \tau_i(U) - \tau_i(U \setminus T)$.*

(Indeed, if $S \subseteq U$, then either $S$ intersects $T$ or $S \subseteq U \setminus T$, implying $\tau_i(U) = \tau_i(U, T) + \tau_i(U \setminus T)$.)

Put together, it suffices to precompue for each node $i$ the values $\tau_i(J)$ for all $J \subseteq C_i$. Since $\tau_i$ is the zeta transform of $\pi_i$ over the subset lattice of $C_i$, it can be computed in time $O(2^K K)$; see Supplement A.1. The space requirement is $O(2^K)$ per node. This improves upon a brute-force approach, which requires building time $O(3^K K^2)$ and storage size $O(3^K)$ per node [19].

When the arithmetic is with fixed-precision numbers, there is a risk of so-called catastrophic cancellation. That is, the outcome of a subtraction may vanish (due to limited precision), even if the exact value is non-zero. While such cases occured only rarely in our experiments, we build a secondary data structure to handle them; if there are $m$ cases, the construction takes $O(3^K n)$ time and $O(3^K + m)$ space (Suppl. A.2). Note: in Table 1 we made the mild assumption that $m = O(2^K n)$.

### 3.3 Markov chain simulation

We follow the partition MCMC method [16, 19] and simulate a Markov chain $R^1, R^2, \ldots, R^L$ of some appropriate length $L$ on ordered set partitions of $V$ using the Metropolis–Hastings algorithm. At state $R^s$ a candidate $R'$ for the next state is generated by either splitting a part, merging two adjacent parts, or swapping nodes in different parts, uniformly at random over the valid choices; denote this distribution by $q(R'|R^s)$. The proposal is accepted as the new state $R^{s+1}$ with probability $\min\{1, \alpha\}$, where $\alpha = \pi(R')/\pi(R^s) \times q(R^s|R')/q(R'|R^s)$; otherwise $R^{s+1}$ is set to $R^s$.

Instead of simulating a single long chain, we enhance the mixing of the chain by employing Metropolis coupling [10]: we run $M > 1$ shorter "heated" chains in parallel, the $k$th chain with stationary distribution proportional to $\pi^{k/M}$. In every other step, two chains $k$ and $l = k + 1$ are selected uniformly at random, and a swap of their states, $R^{s,k}$ and $R^{s,l}$, is proposed; the acceptance ratio $\alpha$ equals the $M$th root of $\pi(R^{s,k})/\pi(R^{s,l})$. In our experiments, we put $M := 16$.

### 3.4 Postprocessing

We generate a DAG per sampled partition as postprocessing, in order to save space. The key observation is that, instead of generating an entire DAG for each partition in turn, we can proceed one node in turn, and generate the parent sets of the node for all the DAGs we are generating. This "transposition trick" enables reusing the space we need for efficient sampling of parent sets. Furthermore, for sampling the parent sets of a fixed node, we introduce a data structure to index certain weighted sums, enabling efficient sampling of constrained sets.

Recall that the root-partition of the DAG in Figure 1(a) is $\{1\}\{2, 3\}\{4\}\{5\}\{6\}$. Now, consider generating a random DAG compatible with this partition. Since each node must take at least one parent from the predecessor part, we must include the edges $5 \to 6$, $4 \to 5$, $1 \to 2$ and $1 \to 3$. In addition, either $2 \to 4$ or $3 \to 4$ is included. The parent sets will be sampled according to the scores $\pi_i$ as explained below such that these restrictions are satisfied.

For a more technical description, consider generating a DAG $G$ from the posterior distribution given that the root-partition of $G$ is $R$. We can draw $G$ by sampling independently for each node $i \in R_t$ a parent set $S \subseteq R_{1,t-1}$ that intersects $R_{t-1}$, with probability proportional to $\pi_i(S)$. If implemented in a direct way, this takes time $O(2^K)$ per node, but no additional space [19].

We reduce the time requirement to $O(K)$, by investing $O(3^K)$ preparation time per node and $O(3^K)$ space in total. Consider a fixed node $i$. The idea is to construct a data structure that, given any node sets $T \subseteq U \subseteq C_i$, enables drawing a parent set $S \subseteq U$ that intersects $T$, with probability proportional to $\pi_i(S)$. We draw $S$ in $O(|U|)$ iterative steps, in each step deciding whether a particular node $j \in U$ is included in $S$ or not. To enable this, our data structure stores the sum of $\pi_i(S)$ over $T' \subseteq S \subseteq U'$ for all pairs $T' \subseteq U' \subseteq C_i$; see Supplement A.3 for details.

If the number of sampled DAGs is $r$, the total space and time requirements of postprocessing are $O(3^K + Knr)$ and $O(3^K n + Knr)$, respectively. In contrast to the brute-force approach [19], our trick makes it feasible to sample large numbers of DAGs.

Table 2: Algorithms for selecting the candidate parents of node $i$

| | |
|---|---|
| *Opt* | Select a $K$-set $C_i$ so as to maximize the posterior probability that $pa(i) \subseteq C_i$ (cf. Prop. 2) |
| *Top* | Select the $K$ nodes $j$ with the highest local score $\pi_i(\{j\})$ |
| *PCb* | Merge the neighborhoods of $i$, excluding children, returned by *PC* on 20 bootstrap samples |
| *MBb* | Merge the Markov blankets of $i$ returned by *IA* on 20 bootstrap samples |
| *GESb* | Merge the neighborhoods of $i$, excluding children, returned by *GES* on 20 bootstrap samples |
| *Greedy* | Iteratively, add a best node to $C_i$, initially empty; the *goodness* of $j$ is $\max_{S \subseteq C_i} \pi_i(S \cup \{j\})$ |
| *Back&Forth* | Starting from a random $K$-set delete a worst and add a best node, alternatingly, until the same |

## 4 Selection of candidate parents

We wish to find a good set of $K$ candidate parents for each node. Our interest is in algorithms that scale up to hundreds of nodes. While we cannot expect an algorithm that always returns an optimal set, we can hope for a heuristic that finds sets covering a large fraction of the graph posterior mass. We formalize this problem, consider the issue of evaluating the performance of a given algorithm, and finally, briefly describe several alternative algorithms and report on an empirical study.

### 4.1 The maximum coverage problem

Consider a tuple of candidate parent sets $C = C_1 C_2 \cdots C_n$. Define the *coverage* of $C$ as the posterior probability that the parents of $i$ belong to $C_i$ for all nodes $i$. Likewise, define the *mean coverage* of $C$ as the average of the marginal posterior probabilities that the parents of $i$ belong to $C_i$.

Given a $C$, we can compute the coverage and mean coverage in time $O(3^n n)$ and space $O(2^n n)$. Namely, within this complexity we can evaluate the partition function [39] as well as the marginal posterior probabilities of all the $2^{n-1}$ possible parent sets of each node [31]. Thus exact evaluation of a given $C$ is computationally feasible up to around $n = 22$.

The *maximum (mean) coverage* problem is to find a $C$ so as to maximize the (mean) coverage, subject to the constraint $|C_i| \leq K$ for all $i$. The mean variant is tractable for small $n$:

**Proposition 2.** *The maximum mean coverage problem can be solved in time $O(3^n n)$.*

*Proof.* Compute first the marginal posterior probabilities $g_i(S) := p(pa(i) = S|X)$ for all $S \subseteq V \setminus \{i\}$ in time $O(3^n n)$ [31]. Then compute $g_i'(T) := \sum_{S \subseteq T} g_i(S)$ for all $T \subseteq V \setminus \{i\}$ in time $O(3^n n)$. Finally, for each $i$ return a $K$-set $C_i$ that maximizes $g_i'(C_i)$; this takes time $O(2^n n)$. $\square$

### 4.2 Scalable algorithms for the maximum coverage problem

For larger numbers of nodes $n$, we have to resort to faster algorithms that are only guaranteed to find a locally optimal collection of candidate parent sets. We tested several heuristics, summarized in Table 2 (details in Suppl. C). Some rely on existing sophisticated algorithms for finding the Markov equivalence class (the *PC* algorithm, using independence tests [35, 5]; greedy equivalence search,

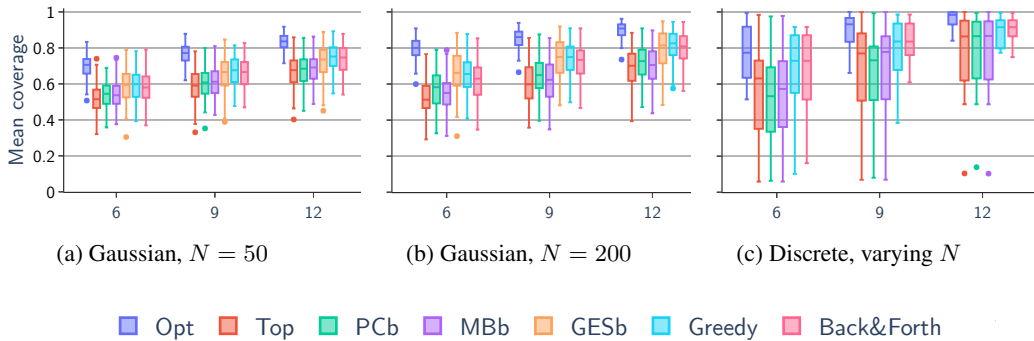

(a) Gaussian, $N = 50$   (b) Gaussian, $N = 200$   (c) Discrete, varying $N$

■ Opt   ■ Top   ■ PCb   ■ MBb   ■ GESb   ■ Greedy   ■ Back&Forth

Figure 2: Performance comparison on selecting $K = 6, 9, 12$ candidate parents with (a, b) synthetic data over 20 nodes and (c) benchmark data sets over 17–23 nodes with $101 \leq N \leq 8124$ data points.

---

**Algorithm 2** The *Beeps* method for sampling from the posterior of linear causal effects

---

1: Sample DAGs $\{G^s\}_{s=1}^L$ approx. from the posterior $p(G|X)$, e.g., using *Gadget* (Section 3).
2: For each $G^s$, sample $B^s$ from the posterior $p(B|G^s, X)$, each row independently (Eq. 2).
3: For each $B^s$, compute the matrix of pairwise causal effects $A^s$ via $A = (I - B)^{-1}$.
4: Output $\{A^s\}_{s=1}^L$.

---

*GES* using the BIC score [4]) or the Markov blanket of a target node (the *Incremental Association* algorithm, *IA* [40]) of the unknown DAG; we ran the basic algorithms on 20 bootstrap samples of the data, took the union of the returned neighborhoods, and removed or added the lowest- or highest-scoring nodes to get exactly $K$ candidates. Other algorithms are more elementary and handle each node separately, considering parent sets that are either singletons or subsets of an already constructed candidate set. Our implementations build on standard software [33, 13, 12, 1].

For an empirical comparison of the heuristics, we set $n$ to 20 to enable exact evaluation of the achieved coverage and comparison to the best possible performance (*Opt*, cf. Prop. 2). We sampled two data sets of size $N = 50$ and $N = 200$ from each of 100 synthetic linear Gaussian DAGs, generated so that the expected neighborhood size was 4, the edge coefficients and the variances of the disturbances uniformly distributed on $\pm[0.1, 2]$ and $[0.5, 2]$, respectively. We observe that the coverage of optimal sets of $K$ candidates increases with $K$ and $N$, reaching 0.90 on average at $K = 12$ and $N = 200$ (Fig. 2(a, b)). *Greedy* is the best of the heuristics and gets the closer to *Opt*, the larger the size $K$.

To investigate the performance on discrete real data, we also ran the algorithms on 8 data sets obtained from the UCI machine learning repository [6], with up to 23 variables, using available preprocessed sets [25]. In Fig. 2(c), we observe that *Greedy* and its *Back&Forth* variant achieve coverages close to *Opt*; the other algorithms perform worse. *GESb* is not shown for discrete data, as the employed software only allowed Gaussian BIC to be used.

## 5 Bayesian estimation of linear causal effects

The ability to sample DAGs (approximately) from the posterior distribution offers us a way to sample (pairwise) causal effects from the posterior distribution in linear Gaussian models. Algorithm 2 outlines our method, dubbed *Beeps* (Bayesian Effect Estimation by Posterior Sampling).

Our goal is to draw a sample from the posterior $p(A|X)$, where $A = (a_{ij})$ is the matrix of pairwise causal effects and $X$ the data. Conveniently, $A$ can be expressed as a converging geometric series w.r.t. the edge weight matrix $B$, resulting in $A = (I - B)^{-1}$. Using this relation, $A$ can readily be computed from samples of $B$ drawn from the posterior $p(B|X)$. To draw $B$, we view $p(B|X)$ as a marginal of $p(B, G|X)$, and by the chain rule, draw first $G$ from $p(G|X)$ and then $B$ from $p(B|G, X)$. In what follows, we assume that $G$ has already been sampled and focus on the latter task.

Recall that we parameterize our linear Gaussian DAG by the mean vector $\boldsymbol{\mu}$, the matrix $B$, and the diagonal matrix of error term precisions $Q$. Geiger and Heckerman [8, 9] showed that there is a unique class of priors over these parameters satisfying the desirable properties of global and local modularity and the score-equivalence of the marginal likelihood $p(X|G)$, the BGe score. Moreover, for any prior from this class, we obtain the posterior of $B$ analytically: the rows of $B$ are independent with a $t$-distribution whose parameters can be efficiently computed. Since some of the key formulas in the literature contain small errors and typos; we give a complete derivation below and in Supplement B.

We begin with a normal–Wishart prior on the parameterization by $\boldsymbol{\mu}$ and the precision matrix $W$:

$$\boldsymbol{\mu} \mid W \sim \mathcal{N}(\boldsymbol{\nu}, \alpha_\mu W), \qquad W \sim \mathcal{W}(T^{-1}, \alpha_w).$$

Here the scalars $\alpha_\mu$, $\alpha_w$, vector $\boldsymbol{\nu}$, and matrix $T$ are hyperparameters, which do not depend on the DAG $G$.[2] By change of variables, this is transformed to a prior over $\boldsymbol{\mu}$, $B$ and $Q$, conditional on $G$ [9]. After integrating out $\boldsymbol{\mu}$, the marginal prior $p(B, Q|G)$ factorizes, due to global and local parameter modularity, into a product of $p(\boldsymbol{b}_i, q_i | pa(i))$ over the nodes $i$; here $\boldsymbol{b}_i$ is the $i$th row of $B$ and $q_i$ the $i$th diagonal element of $Q$. The prior for $\boldsymbol{b}_i$ and $q_i$ given $pa(i)$ is then (see Suppl. B)

$$\boldsymbol{b}_i \mid q_i \sim \mathcal{N}\big((T_{11})^{-1}T_{12}, \, q_i T_{11}\big), \qquad q_i \sim \mathcal{W}\big((T_{22} - T_{21}(T_{11})^{-1}T_{12})^{-1}, \, \alpha_w - n + l\big),$$

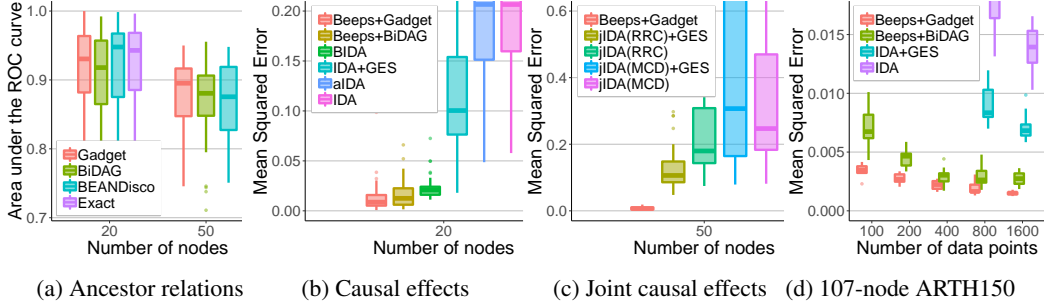

|(a) Ancestor relations|(b) Causal effects|(c) Joint causal effects|(d) 107-node ARTH150|

Figure 3: Performance comparisons. (a) Bayesian methods on inferring ancestor relations from discrete data. Estimating (b) marginal and (c) joint causal effects from Gaussian data. (d) Estimating causal effects from data sets from a benchmark BN. The MCMC methods were ran for 1 and 3 hours for the 20- and 50-node data, respectively, and 12 hours for (d); the other methods were faster.

where the blocks of $T$ are $T_{11} := T[pa(i), pa(i)]$, $T_{12} := (T_{21})^{\mathsf{T}} = T[pa(i), i]$, $T_{22} := T[i, i]$, and $l-1$ is the number of parents of $i$. This corrects some errors in the formulas of Geiger and Heckerman [9] for the degrees of freedom (noted also by Kuipers et al. [17]) and some typos in the matrices.

The posterior is of exactly the same form, just $\alpha_w$ and $T$ replaced, respectively, by

$$\alpha'_w := \alpha_w + N \quad \text{and} \quad R := T + S_N + \frac{\alpha_\mu N}{\alpha_\mu + N}(\boldsymbol{\nu} - \bar{\boldsymbol{x}}_N)(\boldsymbol{\nu} - \bar{\boldsymbol{x}}_N)^{\mathsf{T}},$$

where $\bar{\boldsymbol{x}}_N := \frac{1}{N}\sum_s \boldsymbol{x}_s$ and $S_N = \sum_s (\boldsymbol{x}_s - \bar{\boldsymbol{x}}_N)(\boldsymbol{x}_s - \bar{\boldsymbol{x}}_N)^{\mathsf{T}}$. Finally, integrating out $q_i$ yields

$$\boldsymbol{b}_i \,|\, X, pa(i) \sim t\Big((R_{11})^{-1}R_{12}\,, \ \frac{\alpha'_w - n + l}{R_{22} - R_{21}(R_{11})^{-1}R_{12}}\,R_{11}\,, \ \alpha'_w - n + l\Big). \tag{2}$$

*Beeps* differs from IDA-based methods (including the Bayesian ones, *BIDA* [31] and *OBMA* [2]), which estimate the causal effect from $x$ to $y$ by a linear regression of $y$ on $x$ and the parents of $x$. In contrast, *Beeps* takes into account the whole graph structure and estimates of the single coefficients. This improves accuracy: e.g., the estimate is *always exactly* zero when $x$ is not an ancestor of $y$. Furthermore, our method enables estimation of causal effects under multiple interventions [27] by replacing the coefficients into the intervened variables in $B$ with zero in Step 3 of Algorithm 2.

## 6 Experiments on causal inference

We compared our algorithms[3] to state-of-the-art Bayesian and non-Bayesian methods for discovering ancestor relations and estimating causal effects (marginal and joint). For a complete set of results and the choices of the various user parameters of the methods, we refer to Supplement D.

We first evaluated the efficiency of *Gadget* in sampling DAGs from the posterior and detecting ancestor relations. We considered data on 20 and 50 nodes to enable comparison to an exact algorithm (for $n = 20$) [31] and two state-of-the-art MCMC methods, *BEANDisco* [29] and *BiDAG* [19]. We generated 400 data points from 50 binary BNs with av. neighbourhood size 4. We observe that *Gadget* closely matches or outperforms the other MCMC methods and the exact algorithm (Fig. 3(a)).

We then evaluated the performance of our *Beeps* method in estimating causal effects, using either *Gadget* or *BiDAG* as the DAG sampler. To enable an informative comparison to the state-of-the-art scalable methods, i.e., variants of the IDA method [22], we condense the effect estimates to the mean value and calculate the mean-squared error [31, 2]. We generated 200 data points from 50 Gaussian BNs with neighbourhood size 4. Our method achieves better accuracy in causal effect estimation compared to the BIDA method, which uses exact computation (but a different effect estimation technique (Fig. 3(b)). We evaluated the performance of *Beeps* also in estimating joint causal effects (Fig. 3(c)). Our method clearly outperforms the available IDA-based methods [27] in accuracy.

Finally, we obtained 50 datasets with 100–1600 data points from a benchmark Gaussian BN on gene expressions of Arabidopsis thaliana with $n = 107$ nodes [34, 30]. We ran the MCMC methods 12 hours or up to $10^8$ MCMC iterations. Despite data from a single source, the performance of *BiDAG* varies considerably: for 200-400 data points it can often reach the limiting $10^8$ iterations but for 800 and 1600 data points *BiDAG* fails to complete 100 iterations for $4/10$ and $8/10$ datasets respectively. *Gadget* is able to use $K = 15$ candidate parents for all data sets, and with *Beeps* provide an improved accuracy especially with fewer data points (Fig. 3(d)). See Supplement D for further experiments.

## 7 Concluding remarks

We presented Bayesian methods for discovering causal relations and for estimating linear causal effects from passively observed data. *Gadget* samples DAGs along a Markov chain, building on a recently introduced partition MCMC strategy [16, 19], with several algorithmic modifications to improve the time and memory requirements. We have demonstrated that our method is feasible on systems with one hundred variables, and the theory (Table 1) and simulations suggest that even larger systems, with several hundreds of variables, should be within reach. *Beeps* takes as input a sample of DAGs drawn from a posterior distribution, samples model parameters conditionally on each sampled DAG to obtain a fully specified BN, thereby yielding a sampling-based approximation of the joint posterior of the causal effects; *Beeps* relies on the fact that in a linear model, the effects can be efficiently computed via matrix inversion. A similar sampling-based approach has recently been implemented also for non-linear models with binary variables [26, 18]. However, it requires either computationally expensive exact or approximate inference in the model. Our empirical results on causal effect estimation suggest that Bayesian methods outperform non-Bayesian (IDA-based) ones especially when the data are scarce.

We conclude with two remarks. First, while our data structure for DAG sampling was motivated by a space saving, we may alternatively trade the saving for quick DAG sampling *during the Markov chain simulation*. This would enable a sophisticated edge-reversal move [11], which has proven beneficial in partition MCMC [16] but is not implemented in *BiDAG*, apparently due to its computational cost. Second, we found that *optimal* sets of $K$ candidate parents often yield a good coverage of the posterior with moderate $K$, and that simple heuristics often achieve nearly optimal performance—but not always. The problem warrants further research. E.g., could one here successfully employ techniques that quickly list large numbers of high-scoring parent sets [32]? We believe our approach to compare to optimal sets on moderate-size problem instances should be valuable in the quest.

## Acknowledgments

This work was partially supported by the Academy of Finland, Grant 316771.

## Statement of broader impact

Our work advances computational methods for learning from data. Specifically, we give more efficient and reliable methods for Bayesian statistical inference when the stucture of the underlying graphical model is unknown. A Bayesian posterior is a key enabler in informed and principled risk management and decision making under uncertainty, e.g., via the principle of expected utility; clearly, the concept of causality is essential here. We believe that, in the long run, our work will have broad impact in various areas of other sciences, technology, and in society, by making more efficient use of the available data and incorporating quantifications of uncertainty.

Positive outcomes:

- This work addresses some of the key methodological challenges in computational causal inference, bringing the relatively new field closer towards high-impact real-world applications.
- Shows the advantages of Bayesian inference, inviting and encouraging to use of similar approaches also in other domains.

Negative outcomes:

- Making causal predictions based on observational data is inherently difficult even under rather strong assumptions. Not being aware of these limitations, a non-expert user could potentially overinterpret the results.
- Our methods contribute to the practice of discovering causal and statistical relations from data. There is a risk of biased conclusions if the data are biased (cf. fairness in machine learning).

## Footnotes

[2]With the notation $\alpha_\mu$ and $\alpha_w$ we adhere to the choices in the key references [9, 17].

[3]We provide a Python interface for both algorithms, with many time critical parts implemented in C++. For source code see https://www.cs.helsinki.fi/group/sop/gadget-beeps.

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
