[Supplementary Material]

# Towards Scalable Bayesian Learning of Causal DAGs
## *Supplement*

**Jussi Viinikka**
Department of Computer Science
University of Helsinki
jussi.viinikka@helsinki.fi

**Antti Hyttinen**
HIIT & Department of Computer Science
University of Helsinki
antti.hyttinen@helsinki.fi

**Johan Pensar**
Department of Mathematics
University of Oslo
johanpen@math.uio.no

**Mikko Koivisto**
Department of Computer Science
University of Helsinki
mikko.koivisto@helsinki.fi

## Contents

# A Algorithms

We first recall an algorithm from the literature and then describe in detail our two novel data structures along with associated algorithms for constructing and using them. The last section describes ways to allow a node to take parents also outside the set of candidate parents.

## A.1 Fast zeta transform

We describe a transform that is a basic building block in several of our algorithms (see Sections 3.2 of the main article and Section A.2 below).

Let $U = \{1, 2, \ldots, m\}$. Let $f$ be a function from the subsets of $U$ to real numbers (or, to any ring). The *zeta transform* of $f$ (over the subset lattice of $U$) is the function $g$, defined for all $T \subseteq U$ by

$$g(T) := \sum_{S \subseteq T} f(S) \,.$$

We can evaluate the zeta transform, i.e., compute $g$ given $f$ as input, with $O(2^m m)$ additions [22, 11]. This is achieved by the *fast zeta transform* algorithm, which first puts $g_0 := f$ and then for $i = 1, 2, \ldots, m$ uses the recurrence

$$g_i(T) := g_{i-1}(T \setminus \{i\}) + [i \in T] \, g_{i-1}(T) \,, \qquad T \subseteq U \,,$$

where $[i \in T]$ evaluates to 1 if $i$ belongs to $T$, and to 0 otherwise. One can show that $g_m = g$.

## A.2 Preparing for catastrophic cancellations

Lemma 1 in Section 3.2 of the main article gives us a way to compute any requested score sum by subtracting a smaller sum from a larger sum. We noted that this may result in a catastrophic cancellation due to fixed-precision arithmetic. Here we show how we handle the problematic cases by scanning through them and storing the exact (or, more accurate) values as preprocessing.

Let $i \in V$. Let $T \subseteq U \subseteq V \setminus \{i\}$. Recall that the score sum of interest is defined as

$$\tau_i(U, T) := \sum_{S \subseteq U : S \cap T \neq \emptyset} \pi_i(S) \,.$$

It easy to verify that, for any $j \in T$,

$$\tau_i(U, T) = \tau_i(U, \{j\}) + \tau_i(U \setminus \{j\}, T \setminus \{j\}) \,.$$

In particular, this recurrence holds when $U \subseteq C_i$. Thus, for a fixed $i$, we can compute the values $\tau_i(U, T)$ for all $T \subseteq U \subseteq C_i$ with $O(3^K)$ additions. Observe that the base cases can be written as

$$\tau_i(U, \{j\}) = \sum_{S \subseteq U \setminus \{j\}} \pi_i(S \cup \{j\}) \,,$$

and can thus be computed using fast zeta transform with $O(2^K K^2)$ additions.

To avoid storing all the $n3^K$ numbers, we loop over all $T \subseteq U \subseteq C_i$ and store $\tau_i(U, T)$ if and only if it cannot be reliably computed from the values $\tau_i(U)$ and $\tau_i(U \setminus T)$, that is, if $\tau_i(U) - \tau_i(U \setminus T) \approx 0$ (say, the relative difference is less than $2^{-32}$).

## A.3 Sampling random subsets

Section 3.4 of the main article sketches an efficient technique for sampling a DAG from the posterior conditionally on a given root-partition. The essence of the technique is to construct a data structure for each node $i$ separately so that, given "query" sets $U$ and $T$, we can efficiently generate a parent set $S \subseteq U$ that intersects $T$. Below we describe our technique in more abstract terms of subset sampling.

Let $C$ be a set of $K$ elements. With each subset $X \subseteq C$ associate a weight $w(X) \geq 0$. Consider the problem of generating a random $X$ with probability proportional to $w(X)$ and satisfying $X \subseteq U$ and $X \cap T \neq \emptyset$, where $U \subseteq C$ and $T \subseteq U$ are given sets.

We next give a data structure with the following properties:

- Constructing the data structure takes $O(3^K)$ time and $O(3^K)$ space.
- Sampling a random subset takes $O(K)$ time.

For comparison, a straightforward approach would take either $O(2^{|U|})$ time (linear scan) or $O(4^K)$ space and preprocessing time (preprocessing the queries for all $T$ and $U$).

**Construction**   For all $X \subseteq Y \subseteq C$, define

$$f(X,Y) := \sum_{X \subseteq S \subseteq Y} w(S) \,.$$

Observe that the function $f$ can be computed with $O(3^K)$ additions using the recurrence $f(X,Y) = f(X \cup \{i\}, Y) + f(X, Y \setminus \{i\})$ for any $i \in Y \setminus X$.

**Sampling**   Sample a subset $X$, given $U$ and $T$, as follows. For each $i \in U$ in turn, in an arbitrary order, include $i$ with probability

$$\frac{g(X \cup \{i\}, E)}{g(X, E)} \,, \tag{1}$$

where $g(X, E) := f(X, U \setminus E) - f(X, U \setminus E \setminus T)$ and $X$ and $E$ denote the sets of elements that were already included and excluded, respectively; initially, we set both $X$ and $E$ empty.

To see that $X$ is generated with the correct probability, observe first that the probability of excluding $i$ can be written as

$$\frac{g(X, E \cup \{i\})}{g(X, E)} \,. \tag{2}$$

Namely, a set $S$ contributes to the denominator with weight $w(S)$ exactly when $X \subseteq S \subseteq U \setminus E$ and $S \cap T \neq \emptyset$, and to the numerator in (2) exactly when, in addition, $i \notin S$, and to the numerator in (1) exactly when, in addition, $i \in S$.

Thus, the probability of the decision made in the $t$th round is $g(X_t, E_t)/g(X_{t-1}, E_{t-1})$, where $X_t$ and $E_t$ are the sets of elements included and excluded after the first $t$ rounds. By the chain rule, we get from the telescoping product that $X := X_{|U|}$ is generated with probability $g(X, U \setminus X)/g(\emptyset, \emptyset)$. Observe that $g(X, U \setminus X) = w(X)$ if $X$ intersects $T$, and $g(X, U \setminus X) = 0$ otherwise.

**Confronting catastrophic cancellation**   Due to fixed-precision arithmetic, the computed value of $g(X, E)$ may be zero even if the exact value was non-zero. This approximation may result in an uncontrolled bias in the sampling distribution.

As a remedy, if the computed value $g(X, E)$ has a large relative error (deduced by the terms in the subtraction), we switch over to brute-force sampling, which takes time $O(2^{|U|})$.

### A.4   Allowing parents outside the candidates

Strictly requiring all parents of each node $i$ to come from the established set of $K$ candidate parents $C_i$ has two drawbacks: (i) For some nodes $i$, the found set $C_i$ may not be optimal or sufficiently large to cover the posterior well. (ii) The posterior landscape may contain large zero-probability regions, which makes moving between node partitions inefficient for the Markov chain. A remedy for both issues is to allow any single node $j \neq i$ be a parent of $i$, either in combination with a small number of other arbitrary nodes, like implemented in *Gadget*, or in combination with any number of other parents from the candidates, like implemented in *BiDAG* [13]. Below we present some further details.

**Implementation in *Gadget***   Currently *Gadget* allows a node $i$ to have a parent set that either is contained in the set of candidates $C_i$, or is of size at most $d$, where $d$ is a user parameter, set to 3 in our experiments. Compared to the basic case of $d = 0$, this extension requires some additional work both in preprocessing and in the Markov chain simulation phase.

In preprocessing, we compute the local score for $O(n^d)$ parent sets per node, in addition to the $O(2^K)$ subsets of the candidate set. Adopting previously proposed techniques [5, 17], we sort the parents

sets in decreasing order by the score; this will enable a tolerably fast computation of any queried partial sum of the scores to within a given relative error.

In the simulation phase, when the score sum is needed for a node $i$ over the parent sets that are contained in $U$ and intersect $T$, we scan the sorted list until the accumulated sum is guaranteed to be sufficiently large (we allowed a relative error of $0.1$ in our experiments). Compared to previous implementations of this idea [5, 17], a distinctive feature in our implementation is that we can initiate the accumulating sum by the partial sum contributed by the parent sets that are contained in the candidates; this contribution is often non-zero and expedites the computation.

**Implementation in *BiDAG***    Constant-time score sum computation during simulation is maintained in *BiDAG* by precomputing the score sums for the extended parent sets. This increases the space requirement and the preprocessing time by a factor of $n$, further increasing the gap to the complexity bounds of *Gadget*, necessitating the use of a smaller value of $K$.

Furthermore, a preliminary simulation run is used to extend the initial candidate parent sets (found by the PC algorithm) based on visited high-scoring DAGs [13]. The extension is vital for this procedure.

## B    Bayesian posterior: a derivation

Here we derive formulas for the posterior of the parameters of a linear Gaussian DAG model assuming a normal–Wishart prior. We follow previous similar derivations by Geiger and Heckerman [7] and Kuipers, Moffa, and Heckerman [12]. We refer to the two reference articles by GH and KMH. In more detail, GH derive a prior distribution for the model parameters similar to ours. KMH derives the BGe score, noting errors in the original derivation [6]. We derive here the prior and the posterior of the model parameters, taking into account the KMH corrections and further correcting additional inconsistencies in GH.

### B.1    Prior and posterior distributions with respect to all variables

The basic idea is to first consider a complete DAG. We will specify the prior so that it does not distinguish between equivalent DAGs. Thus, it does not matter which complete DAG we consider. Then, when we proceed to consider a node $i$ in an arbitrary DAG, we can make use of the result we have for any complete DAG that contains the same local pattern, i.e., node $i$ has the same parent set.

We assume $\boldsymbol{x}$ is distributed normally with precision matrix $W$ and mean $\boldsymbol{\mu}$:

$$\boldsymbol{x} \sim \mathcal{N}(\boldsymbol{\mu}, W).$$

Following GH and KMH, $\boldsymbol{\mu}$ and $W$ have a (conjugate) normal–Wishart prior distribution:

$$\boldsymbol{\mu} \sim \mathcal{N}(\boldsymbol{\nu}, \alpha_\mu W), \qquad W \sim \mathcal{W}_n(T^{-1}, \alpha_w)$$

where $\alpha_\mu$ and $\alpha_w$ are equivalent sample sizes, $\boldsymbol{\nu}$ is a mean vector, and $T$ is an inverse scale matrix. Because this is a conjugate prior to normal likelihood, we have that the posterior is of the same form:

$$\boldsymbol{\mu} \sim \mathcal{N}(\boldsymbol{\nu}', \alpha_\mu' W), \qquad W \sim \mathcal{W}_n(R^{-1}, \alpha_w')$$

with updated hyperparameters $\alpha_\mu' := \alpha_\mu + N$, $\alpha_w' := \alpha_w + N$,

$$\boldsymbol{\nu}' := \frac{\alpha_\mu \boldsymbol{\nu} + N \bar{\boldsymbol{x}}_N}{\alpha_\mu + N}, \quad \text{and} \quad R := T + S_N + \frac{\alpha_\mu N}{\alpha_\mu + N}(\boldsymbol{\nu} - \bar{\boldsymbol{x}}_N)(\boldsymbol{\nu} - \bar{\boldsymbol{x}}_N)^\mathsf{T},$$

where $\bar{\boldsymbol{x}}_N = \frac{1}{N}\sum_{s=1}^N \boldsymbol{x}_s$ and $S_N = \sum_{s=1}^N (\boldsymbol{x}_s - \bar{\boldsymbol{x}}_N)(\boldsymbol{x}_s - \bar{\boldsymbol{x}}_N)^\mathsf{T}$.

### B.2    Prior and posterior distributions with respect to subsets of variables

Now consider a node $i$ in an arbitrary DAG. Our goal is to determine the joint posterior distribution of the coefficients $\boldsymbol{b}_i$ associated with the edges from $pa(i)$ to $i$ and the precision of the error term $q_i$. Under the modularity assumption, the local distribution and parameter prior of node $i$ is the same for all DAGs where $pa(i)$ is the parent set of node $i$. In particular, let us consider a complete DAG that has this property and where, in addition, the topological ordering within $pa(i)$ conincides with the

natural ordering of integers. Let $Y := pa(i) \cup \{i\}$, let $l$ be the size of $Y$, and let $Z$ denote the set of remaining $n - l$ nodes. We will need the fact that subgraph induced by $Y$ is complete in the change of parameterization later on. For a vector $\boldsymbol{v}$ and set $S$, we let $\boldsymbol{v}_S$ denote the subvector $(v_j : j \in S)$ where we order the entries in increasing order by $j$, except that $i$ is the last if it belongs to $S$. We use an equivalent notation for submatrices indexed by subsets of rows and columns.

Following KMH, we consider the subvector $\boldsymbol{y} = \boldsymbol{x}_Y$ which has the distribution

$$\boldsymbol{y} \sim \mathcal{N}(\boldsymbol{\mu}_Y, W_Y)$$

where $W_Y := W_{YY} - W_{YZ}(W_{ZZ})^{-1}W_{ZY}$ is obtained by inverting to covariance matrix, marginalizing and inverting back to precision. The prior on $\boldsymbol{\mu}_Y$ and $W_Y$ is

$$\boldsymbol{\mu}_Y \sim \mathcal{N}(\boldsymbol{\nu}_Y, \alpha_\mu W_Y), \qquad W_Y \sim \mathcal{W}_l\left((T_{YY})^{-1}, \alpha_w - n + l\right).$$

This result is obtained in Equation A.24 in KMH. To get to the posterior, we can make a similar transformation of the full posterior to this subset:

$$\boldsymbol{\mu}_Y \sim \mathcal{N}(\boldsymbol{\nu}'_Y, \alpha'_\mu W_Y), \qquad W_Y \sim \mathcal{W}_l\left((R_{YY})^{-1}, \alpha'_w - n + l\right).$$

Note that the degrees of freedom in the above Wishart distribution has been reduced compared to the corresponding distribution in the previous subsection. This result is in Equation A.26 in KMH.

## B.3 Change of parameterization

We have that $B_{YY}$ is a full lower triangular $l \times l$ matrix with $(\boldsymbol{b}_i, 0)$ as the $l$th row. Likewise, $Q_{YY}$ is an $l \times l$ diagonal matrix including the precisions of the error terms with $q_i$ as the last element. We will utilize the structural equation model

$$\boldsymbol{y} := \boldsymbol{\mu}_Y + B_{YY}(\boldsymbol{y} - \boldsymbol{\mu}_Y) + \boldsymbol{e}_Y,$$

from which we can solve for $\boldsymbol{y}$:

$$\boldsymbol{y} = \boldsymbol{\mu}_Y + (I - B_{YY})^{-1}\boldsymbol{e}_Y.$$

Now we change the parameterization from $W_Y$ to $(B_{YY}, Q_{YY})$ using a bijective transformation $f$. Following Gelman et al. [8, p. 21–22], the density function of $(B_{YY}, Q_{YY})$ is obtained as

$$p(B_{YY}, Q_{YY}) = |\det J| \cdot p(f^{-1}(B_{YY}, Q_{YY}))$$

where

$$f^{-1}(B_{YY}, Q_{YY}) = (I - B_{YY})^\mathsf{T} Q_{YY}(I - B_{YY}),$$

and $J$ is the Jacobian matrix, i.e., the square matrix of partial derivates of $f^{-1}$; $f$ is one-to-one, since $B_{YY}$ corresponds to a full DAG. The matrix $W_Y$ can be represented in a block form as follows: Denote by $B_{11}$ (resp. $Q_{11}$) the submatrix of $B_{YY}$ (resp. $Q_{YY}$) where the last row and the last column are removed. Now we have

$$
\begin{aligned}
W_Y &= \begin{bmatrix} (I - B_{11})^\mathsf{T} & -\boldsymbol{b}_i \\ 0 & 1 \end{bmatrix} \begin{bmatrix} Q_{11} & 0 \\ 0 & q_i \end{bmatrix} \begin{bmatrix} I - B_{11} & 0 \\ -\boldsymbol{b}_i{}^\mathsf{T} & 1 \end{bmatrix} \\
&= \begin{bmatrix} (I - B_{11})^\mathsf{T} Q_{11} & -\boldsymbol{b}_i q_i \\ 0 & q_i \end{bmatrix} \begin{bmatrix} I - B_{11} & 0 \\ -\boldsymbol{b}_i{}^\mathsf{T} & 1 \end{bmatrix} \\
&= \begin{bmatrix} (I - B_{11})^\mathsf{T} Q_{11}(I - B_{11}) + q_i \boldsymbol{b}_i \boldsymbol{b}_i{}^\mathsf{T} & -q_i \boldsymbol{b}_i \\ -q_i \boldsymbol{b}_i{}^\mathsf{T} & q_i \end{bmatrix}.
\end{aligned}
$$

The absolute value of the Jacobian determinant can be obtained by direct calculation using a similar recursion as in Theorem 6 of Geiger and Heckerman [6]:

$$|\det J| = \prod_{j \in Y} q_j^{k_j - 1} \tag{3}$$

where $k_j$ is the index of the node $j$ in $Y$. Note that this product contains the factor $q_i^{l-1}$.

## B.4 Posterior of the coefficients and the precision

Following KMH, the posterior density function of the $k$-dimensional Wishart distribution is

$$\mathcal{W}_k(W|T^{-1},\alpha_w) \quad = \quad \frac{|W|^{(\alpha_w-k-1)/2}}{Z_W(k,T,\alpha_w)}\exp\Big\{-\frac{1}{2}\operatorname{tr}(TW)\Big\}, \tag{4}$$

where $Z_W(k,T,\alpha_w)$ is the normalizing constant and $W$ is positive definite. Plugging in the parameters $T:=R_{YY}$, $W:=W_Y$, $k:=l$, and $\alpha_w := \alpha_w' - n + l$ yields

$$\mathcal{W}_l\big(W_Y|(R_{YY})^{-1},\,\alpha_w'-n+l\big) \quad = \quad \frac{|W_Y|^{(\alpha_w'-n-1)/2}}{Z_W(l,R_{YY},\alpha_w'-n+l)}\exp\Big\{-\frac{1}{2}\operatorname{tr}(R_{YY}W_Y)\Big\}.$$

The change of parametrization then gives us the posterior density

$$\pi(B_{YY},Q_{YY}) \quad \propto \quad \Big(\prod_{j\in Y}q_j^{k_j-1}\Big)\big|(I-B_{YY})^\mathsf{T}Q_{YY}(I-B_{YY})\big|^{(\alpha_w'-n-1)/2}$$

$$\times \exp\Big\{-\frac{1}{2}\operatorname{tr}\big(R_{YY}(I-B_{YY})^\mathsf{T}Q_{YY}(I-B_{YY})\big)\Big\}.$$

The trace term in the exponent can be calculated by blocks:

$$\operatorname{tr}\big(R_{YY}(I-B_{YY})^\mathsf{T}Q_{YY}(I-B_{YY})\big)$$

$$= \quad \operatorname{tr}\left(\begin{bmatrix} R_{11} & R_{12} \\ R_{21} & R_{22} \end{bmatrix}\begin{bmatrix} (I-B_{11})^\mathsf{T}Q_{11}(I-B_{11})+q_i\boldsymbol{b}_i\boldsymbol{b}_i^\mathsf{T} & -q_i\boldsymbol{b}_i \\ -q_i\boldsymbol{b}_i^\mathsf{T} & q_i \end{bmatrix}\right)$$

$$= \quad \operatorname{tr}\big(R_{11}(I-B_{11})^\mathsf{T}Q_{11}(I-B_{11})+q_iR_{11}\boldsymbol{b}_i\boldsymbol{b}_i^\mathsf{T}-q_iR_{12}\boldsymbol{b}_i^\mathsf{T}\big)+R_{22}q_i-q_iR_{21}\boldsymbol{b}_i$$

$$= \quad q_i\boldsymbol{b}_i^\mathsf{T}R_{11}\boldsymbol{b}_i-2q_iR_{21}\boldsymbol{b}_i+R_{22}q_i+c$$

$$= \quad q_i\boldsymbol{b}_i^\mathsf{T}R_{11}\boldsymbol{b}_i-2q_iR_{21}(R_{11})^{-\mathsf{T}}R_{11}\boldsymbol{b}_i+q_iR_{21}(R_{11})^{-\mathsf{T}}R_{11}(R_{11})^{-1}R_{12}$$

$$\qquad - q_iR_{21}(R_{11})^{-\mathsf{T}}R_{11}(R_{11})^{-1}R_{12}+R_{22}q_i+c$$

$$= \quad q_i\big(\boldsymbol{b}_i-(R_{11})^{-1}R_{12}\big)^\mathsf{T}R_{11}\big(\boldsymbol{b}_i-(R_{11})^{-1}R_{12}\big)+q_i\big(R_{22}-R_{21}(R_{11})^{-1}R_{12}\big)+c,\tag{5}$$

where $c$ collects any terms that are constant with respect to $\boldsymbol{b}_i$ and $q_i$.

The determinant term simplifies since the determinant of a triangular matrix is the product of its diagonal entries:

$$\big|(I-B_{YY})^\mathsf{T}Q_{YY}(I-B_{YY})\big|^{(\alpha_w'-n-1)/2} = |Q_{YY}|^{(\alpha_w'-n-1)/2} = \prod_{j\in Y}q_j^{(\alpha_w'-n-1)/2}.\tag{6}$$

Putting together the exponent (Eq. 5), determinant (Eq. 6), and the Jacobian (Eq. 3) gives

$$\pi(\boldsymbol{b}_i,q_i) \propto q_i^{l-1}q_i^{(\alpha_w'-n-1)/2}\exp\Big\{-\frac{1}{2}\Big[q_i\big(\boldsymbol{b}_i-(R_{11})^{-1}R_{12}\big)^\mathsf{T}R_{11}\big(\boldsymbol{b}_i-(R_{11})^{-1}R_{12}\big)$$

$$+ q_i\big(R_{22}-R_{21}(R_{11})^{-1}R_{12}\big)\Big]\Big\}.\tag{7}$$

The first term in the exponent implies that:

$$\boldsymbol{b}_i\mid q_i \sim \mathcal{N}\big((R_{11})^{-1}R_{12},\,q_iR_{11}\big).$$

The normalizing constant for the normal distribution includes the term

$$\big|(q_iR_{11})^{-1}\big|^{-1/2} \quad = \quad \big|(q_iR_{11})\big|^{1/2} \propto q_i^{(l-1)/2}.$$

Thus, marginalizing out $\boldsymbol{b}_i$ leaves

$$\pi(q_i) \propto q_i^{(l-1)/2}q_i^{(\alpha_w'-n-1)/2}\exp\Big\{-\frac{1}{2}q_i\big(R_{22}-R_{21}(R_{11})^{-1}R_{12}\big)\Big\}.$$

This is a one-dimensional Wishart (Gamma) distribution, see Equation 4. Thus

$$q_i \sim \mathcal{W}_1\Big(\big(R_{22}-R_{21}(R_{11})^{-1}R_{12}\big)^{-1},\,\alpha_w'-n+l\Big).$$

## B.5 Prior of the coefficients and the precision

If we replace $R$ with $T$ and $\alpha'_w$ with $\alpha_w$ in the above derivation, we can obtain the priors:

$$\boldsymbol{b}_i \mid q_i \sim \mathcal{N}\left((T_{11})^{-1}T_{12}, \, q_i T_{11}\right), \qquad q_i \sim \mathcal{W}_1\left((T_{22} - T_{21}(T_{11})^{-1}T_{12})^{-1}, \, \alpha_w - n + l\right).$$

Compared to p. 1425 in GH the precision/covariance of $\boldsymbol{b}_i$ is different. The dimensions obtained in our derivation correspond to the dimensions of $\boldsymbol{b}_i$ correctly. Furthermore, the degrees of freedom differ; ours take into account the change due to considering subset of variables, pointed out by KMH.

## B.6 Marginal posterior of the edge coefficients

We can still integrate out $q_i$ to get the marginal posterior of $\boldsymbol{b}_i$. The non-constant terms in the joint density in Equation 7 are

$$q_i^{(\alpha'_w - n + 2l - 3)/2} \exp\left\{ -\frac{1}{2}\left[ (\boldsymbol{b}_i - (R_{11})^{-1}R_{12})^{\mathsf{T}} R_{11} (\boldsymbol{b}_i - (R_{11})^{-1}R_{12}) \right.\right.$$
$$\left.\left. + (R_{22} - R_{21}(R_{11})^{-1}R_{12}) \right] q_i \right\}.$$

Integrating this over $q_i$ results in a Gamma integral, which evaluates to

$$\Gamma((\alpha'_w - n + 2l - 1)/2) \left\{ \frac{1}{2}\left[ (\boldsymbol{b}_i - (R_{11})^{-1}R_{12})^{\mathsf{T}} R_{11} (\boldsymbol{b}_i - (R_{11})^{-1}R_{12}) \right.\right.$$
$$\left.\left. + (R_{22} - R_{21}(R_{11})^{-1}R_{12}) \right] \right\}^{-(\alpha'_w - n + 2l - 1)/2}.$$

This implies that

$$\pi(\boldsymbol{b}_i) \propto \left( 1 + \frac{1}{\alpha'_w - n + l} (\boldsymbol{b}_i - (R_{11})^{-1}R_{12})^{\mathsf{T}} \frac{\alpha'_w - n + l}{R_{22} - R_{21}(R_{11})^{-1}R_{12}} \right.$$
$$\left. \times R_{11}(\boldsymbol{b}_i - (R_{11})^{-1}R_{12}) \right)^{-(\alpha'_w - n + l + l - 1)/2},$$

and since $\boldsymbol{b}_i$ has $l - 1$ elements, we have that (see, e.g., Gelman et al. [8])

$$\boldsymbol{b}_i \sim t_{l-1}\left( (R_{11})^{-1}R_{12}, \, \frac{\alpha'_w - n + l}{R_{22} - R_{21}(R_{11})^{-1}R_{12}} R_{11}, \, \alpha'_w - n + l \right),$$

where the middle term marks precision.

# C  Candidate parent selection

Here we describe in detail the algorithms used for selecting the $K$ candidate parents and how the performance of the algorithms was evaluated. Some implementation practicalities are also discussed.

## C.1 Optimal algorithm and heuristics

Unless otherwise specified, the local scores referred to in the following are as specified in section D.1.

*Opt*  The *Opt* (i.e., optimal) algorithm selects a $K$-set $C_i$ so as to maximize the posterior probability that $pa(i) \subseteq C_i$ (cf. Proposition 2 in the main paper). First all the local scores are computed, after which the (unnormalized) parent set probabilities are computed by summing for each node and parent set the scores of DAGs where the variable has the given parent set. As a last step, for each node $i$ and subset of nodes $C_i \subseteq V \setminus \{i\}$ of size $K$ (that is, for each possible set of candidate parents) the sum of probabilities over the subsets of $C_i$ is computed, and the set maximizing the sum is finally output. Note that *Opt* is scalable only up to around 25 variables and we use it here for a reference.

*Top*  Select the $K$ nodes $j$ with the highest local score $\pi_i(\{j\})$. The heuristic therefore only considers parent sets of size 1 and will miss any candidate parent whose value is dependent on being in a set with a number of others.

*PCb* Merge the neighbourhoods of $i$, excluding children, returned by *PC* on 20 bootstrap samples. The parameters of the algorithm are the p-value threshold and the maximum conditioning set size in the independence tests. To avoid being overly conservative, we set the p-value to $0.10$ and the maximum conditioning set size to 1.

*MBb* Merge the Markov blankets of $i$ returned by the incremental association (*IA*) algorithm on 20 bootstrap samples. The p-value threshold and maximum conditioning set size were set to the same values as in *PCb*.

*GESb* Merge the neighborhoods of $i$, excluding children, returned by greedy equivalence search (*GES*) on 20 bootstrap samples. The score function used (in pcalg [10, 9]) is BIC.

*Greedy* Iteratively, add a best node to $C_i$, initially empty, where *goodness* of $j$ is $\max_{S \subseteq C_i} \pi_i(S \cup \{j\})$. That is, add the node with which we can get the next highest uncovered local score covered.

*Back&Forth* Using the definition of goodness from *Greedy*, start from a random $K$-set, delete a worst and add a best node, alternatingly, until the added node is the one deleted in the previous step.

*Greedy-lite* A computationally more efficient variant of *Greedy*. First, build a candidate set $C_i$ of $K - s$ nodes with *Greedy*. Then, instead of adding the single best node, add the $s$ best nodes in a single step, where the goodness of a node is defined as in *Greedy*. We set $s := 6$ to limit the number of scores that we have to compute by a factor of $2^6 = 64$, as compared to *Greedy*.

*Gadget*, including the candidate selection phase, in its current version is implemented mostly in Python, with some time critical parts in C++. The local scores are computed with the Python version of Gobnilp [1, 4]. The PC and Incremental Association algorithms are implemented in the bnlearn R-package [20], which our code interfaces with. Similarly, we use GES as implemented in the pcalg R-package [10, 9]. To compute the marginal posterior parent set probabilities (as per Proposition 2 in the main paper), allowing both for the evaluation of the heuristics and for computing the optimal parent sets when $n$ is small, we use software developed by Pensar et al. [19].

As the algorithms *PCb*, *MBb* and *GESb* as described can return any number of candidate parents for each node, there has to be a mechanism for adjusting the number to match the desired $K$ exactly. In our experiments we tried two approaches: adding (removing) nodes randomly, or in the order given by the scores of their singular parent sets. The latter proved more performant and was therefore used. Consequently, the selection of the parameters for the used PC and IA algorithms also determines how closely the returned candidates mirror those of *Top* – if the initial phase of the heuristics return an empty graph, the candidates finally returned equal those given by *Top*. Bootstrapping the input data has a similar effect, as it can only increase the number of candidates returned by the initial parts of the heuristics.

In terms of speed, the scoring code we use does not seem particularly well suited to *Opt*, or the heuristics *Greedy* and *Back&Forth*, which require large numbers of scores to be computed. On the other hand, *Greedy* seems like a good candidate as the default algorithm, as it achieves the greatest coverage of the posterior mass for sufficiently large $K$, close to that of the reference *Opt*. *Back&Forth* possibly offers only a slight advantage over *Greedy* in some cases (Figure 2 in main paper; Figure C.2). Thus, in order to avoid the candidate selection phase dominating the time use in the MCMC runs, while still constructing candidate sets that cover close to equal amount of the posterior mass as those constructed by *Greedy*, we used the more efficient *Greedy-lite* variant of it for the main experiments in the paper.

Apart from the heuristics listed, we also experimented with numerous others. These included, for example, hybrid ones which ran a number of heuristics in parallel for increasing $K' \leq K$ until the size of the union of the candidates they found reached the target $K$. The results, however, did not show marked improvement over the simpler methods.

## C.2 Test data

Gaussian data was generated as explained in section 4.2 of the main paper.

(a) Gaussian, $n = 20$, $N = 50$        (b) Gaussian, $n = 20$, $N = 200$

☐ Opt   ☐ Top   ☐ PCb   ☐ MBb   ☐ GESb   ☐ Greedy   ☐ Back&Forth

Figure C.1: Distribution of coverages over the $n$ nodes for 4 randomly selected Gaussian datasets.

For the experiments on discrete data, we used the UCI data sets utilized by Malone et al. [15] for learning discrete Bayesian networks. In the paper we included all the data sets with up to 23 variables, to allow for exact evaluation of the parent set probabilities[1].

## C.3    Empirical results

To evaluate the returned candidates for a given node, when the number of variables is sufficiently small to allow for computing the exact parent set posteriors, we simply summed over all the posteriors of the subsets of the candidates. Finally, we reported the mean over the nodes of the *coverages* thus obtained (Figure 2 in the main paper). Here we break down the analysis further by considering the distribution of the posterior mass covered by the candidate parents of each node. The results in Figures C.1 and C.2 indicate, apart from the variation between different data sets, that even when a heuristic performs well on average there are often nodes for which the candidate parents do not cover a proportionate part of the posterior mass (e.g., Figure C.2(f)).

# D    DAG sampling and causal effect estimation

Here we describe in detail the algorithms used for estimating causal effect and discovering ancestor relations. We also describe how the data was generated and how the performance of the algorithms was evaluated. We present further results and discuss some implementation practicalities.

## D.1    Tested methods

We first describe the hyperparameters and implementation particulars of our novel methods, and then previous methods. We also present further simulation results.

**Hyperparameters of Bayesian models**    Unless noted otherwise, we set the hyperparameters of the priors as follows. For continuous data we use BGe (i.e., a normal–Wishart prior) with $\alpha_\mu = 1$, $\alpha_w = n + 2$, and $T = \frac{1}{2}I_n$ as default in Gobnilp [4]. For discrete data we employ BDeu with equivalent sample size 10. As described in Section 2 of the main paper, we set the prior probability of a DAG proportional to $1/\prod_{i=1}^{n} \binom{n-1}{d_i}$, where $d_i$ is the number of parents of node $i$ in the DAG. These choices ensure that Markov equivalent DAGs receive the same score, i.e., posterior probability; while we regard this result as a folklore, we include the following proof for completeness:

**Proposition D.1.** *The multiset of node indegrees is unique for DAGs in the same equivalence class.*

*Proof.* An edge $i \to j$ is called *covered* in a DAG $G$ if $\mathrm{pa}_G(j) = \mathrm{pa}_G(i) \cup \{i\}$ [3, Def. 2]. Consider reversing $i \to j$ to $i \leftarrow j$ in $G$ to form $G'$. Because a covered edge cannot be a part of an unshielded v-structure, we have that $G$ and $G'$ are in the same Markov equivalence class. Furthermore, a covered edge reversal does not change the multiset of node indegrees, since the indegrees of nodes $i$ and

(a) VOTING, $n = 17$, $N = 435$   (b) ZOO, $n = 17$, $N = 101$   (c) LYMPH, $n = 18$, $N = 148$

(d) EUCALYPTUS, $n = 20$, $N = 736$   (e) HEPATITIS, $n = 20$, $N = 155$   (f) CREDIT-G, $n = 21$, $N = 1000$

(g) HYPOTHYROID, $n = 22$, $N = 3772$   (h) MUSHROOM, $n = 22$, $N = 8124$   (i) SPECT, $n = 23$, $N = 267$

Opt    Top    PCb    MBb    Greedy    Back&Forth

Figure C.2: Distribution of coverages over the $n$ nodes for each included UCI data set.

$j$ are simply switched. Now, by Theorem 2 of Chickering [3], one can move through all DAGs in a Markov equivalence class by a sequence of covered edge reversals. Hence, since the multiset of node indegrees remains unaltered in any single covered edge reversal, all members of a Markov equivalence class must have the same multiset of node indegrees.  □

## Our methods

*Gadget* For selecting candidate parents, *Gadget* uses *Greedy-lite*. The number of candidate parents $K$ was set as large as possible such that computations other than MCMC iterations took at most a half of the allowed time budget. The running time performance of the different parts of the system was estimated for each input by a short preliminary test run. The first 50 % of the iterations were disregarded as burn-in, and thinning was set to obtain 10 000 DAGs.

*Beeps* This essentially implements Algorithm 2 of the main paper in R. *Beeps* can utilize DAGs sampled by either *Gadget* or *BiDAG*. The employed normal–Wishart prior is the same as used for sampling DAGs.

## Previous methods for averaging over DAGs

*BiDAG* We use the `partitionMCMC` function implemented in the BiDAG R-package [13]. The algorithm determines its sampling space and the number of candidate parents automatically using a search. In some of the 107-variable runs, `partitionMCMC` exits and suggests to increase `HARDLIMIT` of the allowed number of possible parents ($K$). In these cases we reran `partitionMCMC` with an increased `HARDLIMIT`. The first 50 %

(a) 46-node ECOLI70　　　(b) 44-node MAGIC-NIAB　　　(c) 64-node MAGIC-IRRI

(d) Ancestor relations for Gaussian data　　　(e) The effect of different p-value thresholds on *IDA* and *aIDA*

Figure D.1: Further performance comparisons. The MCMC methods were run for 1 hours for 20-node data and 3 hours for the benchmarks.

of the iterations were disregarded as burn-in, and thinning was set such that we obtain 10 000 DAGs.

*BEANDisco* We use the authors' implementation available online [18]. For 20 and 50 variables, the maximum number of parents was set to 5 and 4, respectively. Note that *BEANDisco* employs a so-called order modular graph prior, which results in a posterior that is not score equivalent. The first 50 % of the iterations were disregarded as burn-in.

*Exact* This is the ARP algorithm of Pensar et al. [19], which computes the exact posterior of ancestor relations using dynamic programming and inclusion–exclusion recurrences.

**Methods based on IDA**

*BIDA* For BIDA we use the available original implementation with the default priors and scores mentioned in the paper [19]. In particular, they employ a fractional marginal likelihood based score.

*IDA* We use IDA and PC from the pcalg package [10]. The p-value threshold is set to 0.05. Figure D.1(e) shows that the threshold does not have a major effect on the accuracy for the considered datasets.

*IDA+GES* We use IDA from the pcalg package [10, 14]. We employ the GES algorithm in combination with the BIC score, also from pcalg [10]. IDA has been previously coupled with GES [2] and other structure learning algorithms [19].

*aIDA* We use aIDA with the default settings suggested in the implementation [21], thus setting p-value threshold of the PC algorithm to 0.1. Figure D.1(e) shows that the threshold does not have a major effect on the accuracy for the considered datasets.

*jIDA* We test both methods, RRC and MCD, as implemented in the pcalg package [10, 16]. We employ PC with a p-value threshold 0.05 and GES with BIC for obtaining the Markov equivalence class.

Unfortunately, we were not able to get sensible results from the R-code accompanying Castelletti and Consonni [2] for the data set sizes considered here.

(a) Marginal causal effects for Gaussian data.    (b) Ancestor relations for Gaussian data.

Figure D.2: The effect of the number of candidate parents ($K$) on estimation performance. The MCMC methods were run for 1 hours for 20-node data and 3 hours for the 50-node data.

## D.2  Test data

For the synthetic models, edges were included in the graph randomly such that the average neighbourhood size was 4. The linear Gaussian data were generated as described in the main paper Section 4.2, and standardized to zero mean and unit variance [14, Assumption B]. The true causal effects were calculated from the standardized models. For the discrete case, we considered binary variables and the model parameters were drawn from a Dirichlet with an equivalent sample size (ESS) of 10.

## D.3  Empirical results

**Marginal causal effects**    Figures D.1(a–c) show the performance of Gadget and the IDA-based methods in estimating causal effects for additional benchmark datasets obtained from the BNLEARN-network repository [20]. *Beeps+Gadget*, with the running time of 3 hours, is able to provide more accurate estimates, and the accuracy is improved with increasing number of data points. *IDA+GES* needs twice as many data points to reach a similar level of accuracy as *Beeps+Gadget*. Figure D.1(e) shows that different p-value thresholds do not improve the performance of the methods employing the PC algorithm.

**Ancestor relations**    Figure D.1(d) shows that all MCMC methods are able to closely match the performance of the exact approach in detecting ancestor relations in linear Gaussian data. This is similar behaviour as seen in Figure 3(a) in the main paper for discrete data. Note that ancestor relation posteriors are more accurate in predicting the presence of ancestor relations in the true graph, than various applied IDA-based approaches [19].

**Joint causal effects**    In Figure 3(c) in the main paper we evaluated the quality of the estimated causal effects under multiple interventions. We plot the MSE of the estimated causal effects w.r.t. the true ones, where all successive pairs of variables (i.e., $\{x_1, x_2\}, \{x_2, x_3\}, \ldots, \{x_{n-1}, x_n\}$) are intervened on and we consider all causal effects of the intervened variables on the remaining variables.

**The effect of the number of candidate parents ($K$)**    Figure D.2 shows the accuracy performance of *Beeps+Gadget* when using different values for the number of candidate parents $K$. These results indicate that larger $K$ values generally produce better accuracy performance. For 50 variable the higher $K$ values mean shorter MCMC chains are possible within the time budget of 3 hours—this results in a slight drop in accuracy for detecting ancestor relations in Figure D.2(b).

**Mixing of Markov chains**    Figure D.3 shows the mixing performance of *Gadget* on datasets of 100 and 1600 data points sampled from two benchmark Gaussian BNs from the BNLEARN-network repository [20]. The networks, ECOLI70 and ARTH150, specify a distribution on 46 and 107 nodes, respectively. The running times were 3 hours for ECOLI70 (as in Figure D.1), and 12 hours for ARTH150 (as in Figure 3 in the main paper). The 7 independent runs for each data set reach similar levels of posterior probability, with similar variance, indicating good mixing performance.

**Running time performance**    Table D.1 reports example running times for the different parts of our methods. Most time is spend in pre-computation, and in sampling root-partitions and DAGs.

Figure D.3: Mixing of Gadget on data sampled from two benchmark BNs. $Y$-axis shows the posterior probability of the sampled DAGs (a logarithm of the unnormalized posterior). $X$-axis represents the simulation steps after the burn-in phase, during which 100 evenly spaced DAGs were sampled. The columns show the results for 7 independent runs. Burn-in was set to $50\%$ of the chain length, and the running time was set to 3 hours for ECOLI70 and 12 hours for ARTH150.

**Infrastructure**    The experiments were run in computer clusters employing Intel Xeon E5-2680 v4 processors.

## Footnotes

[1]We did not include LETTER ($n = 17$, $N = 20\,000$), however, which proved to be too difficult to compute all local scores for with our setup, presumably due to the large number of data points.

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

Table D.1: Examples of the share of running time among the components of *Gadget* and *Beeps*. Note that the running times spent on different components depend on which $K$ and number of iterations are used. The discrete data runs did not include any computations for causal effects.

| Data type | Gaussian | Discrete | Gaussian |
|---|---|---|---|
| Number of nodes $n$ | 20 | 50 | 107 |
| Number of candidate parents $K$ | 14 | 14 | 15 |
| Search for candidate parents | $< 1\,\%$ | $< 1\,\%$ | $1\,\%$ |
| Precomputing for MCMC | $3\,\%$ | $3\,\%$ | $9\,\%$ |
| MCMC iterations | $79\,\%$ | $86\,\%$ | $69\,\%$ |
| Precomputing for DAG sampling | $14\,\%$ | $9\,\%$ | $17\,\%$ |
| DAG sampling | $2\,\%$ | $2\,\%$ | $4\,\%$ |
| Computing causal effects | $1\,\%$ | – | $< 1\,\%$ |
| Total time | 1 h | 3 h | 12 h |

[3] David Maxwell Chickering. A transformational characterization of equivalent Bayesian network structures. In *Proceedings of the Eleventh Annual Conference on Uncertainty in Artificial Intelligence, UAI 1995*, pages 87–98. Morgan Kaufmann, 1995.

[4] James Cussens. Bayesian network learning with cutting planes. In *The Twenty-Seventh Conference on Uncertainty in Artificial Intelligence, UAI 2011*, pages 153–160. AUAI Press, 2011.

[5] Nir Friedman and Daphne Koller. Being Bayesian about network structure. A Bayesian approach to structure discovery in Bayesian networks. *Machine Learning*, 50(1-2):95–125, 2003.

[6] Dan Geiger and David Heckerman. Learning Gaussian networks. In *The Tenth International Conference on Uncertainty in Artificial Intelligence, UAI 1994*, pages 235–243. Morgan Kaufmann Publishers Inc., 1994.

[7] Dan Geiger and David Heckerman. Parameter priors for directed acyclic graphical models and the characterization of several probability distributions. *The Annals of Statistics*, 30(5):1412–1440, 2002.

[8] Andrew Gelman, Hal S Stern, John B Carlin, David B Dunson, Aki Vehtari, and Donald B Rubin. *Bayesian data analysis*. Chapman and Hall/CRC, third edition, 2014.

[9] Alain Hauser and Peter Bühlmann. Characterization and greedy learning of interventional Markov equivalence classes of directed acyclic graphs. *Journal of Machine Learning Research*, 13:2409–2464, 2012.

[10] Markus Kalisch, Martin Mächler, Diego Colombo, Marloes H. Maathuis, and Peter Bühlmann. Causal inference using graphical models with the R package pcalg. *Journal of Statistical Software*, 47(11):1–26, 2012.

[11] Robert Kennes. Computational aspects of the Möbius transformation of graphs. *IEEE Transactions on Systems, Man and Cybernetics*, 22(2):201–223, 1992.

[12] Jack Kuipers, Giusi Moffa, and David Heckerman. Addendum on the scoring of Gaussian directed acyclic graphical models. *The Annals of Statistics*, 42(4):1689–1691, 2014.

[13] Jack Kuipers, Polina Suter, and Giusi Moffa. Efficient sampling and structure learning of Bayesian networks, 2020. arXiv:1803.07859v3 [stat.ML].

[14] Marloes H. Maathuis, Markus Kalisch, and Peter Bühlmann. Estimating high-dimensional intervention effects from observational data. *The Annals of Statistics*, 37(6A):3133–3164, 2009.

[15] Brandon Malone, Kustaa Kangas, Matti Järvisalo, Mikko Koivisto, and Petri Myllymäki. Empirical hardness of finding optimal Bayesian network structures: Algorithm selection and runtime prediction. *Machine Learning*, 107(1):247–283, 2018.

[16] Preetam Nandy, Marloes H. Maathuis, and Thomas S. Richardson. Estimating the effect of joint interventions from observational data in sparse high-dimensional settings. *The Annals of Statistics*, 45(2):647–674, 2017.

[17] Teppo Niinimäki and Mikko Koivisto. Treedy: A heuristic for counting and sampling subsets. In *The Twenty-Ninth Conference on Uncertainty in Artificial Intelligence, UAI 2013*, pages 469–477. AUAI Press, 2013.

[18] Teppo Niinimäki, Pekka Parviainen, and Mikko Koivisto. Structure discovery in Bayesian networks by sampling partial orders. *Journal of Machine Learning Research*, 17:1–47, 2016.

[19] Johan Pensar, Topi Talvitie, Antti Hyttinen, and Mikko Koivisto. A Bayesian approach for estimating causal effects from observational data. In *The Thirty-Fourth AAAI Conference on Artificial Intelligence, AAAI 2020*. AAAI Press, 2020.

[20] Marco Scutari. Learning Bayesian networks with the bnlearn R package. *Journal of Statistical Software*, 35(3):1–22, 2010.

[21] Franziska Taruttis, Rainer Spang, and Julia C. Engelmann. A statistical approach to virtual cellular experiments: improved causal discovery using accumulation IDA (aIDA). *Bioinformatics*, 31(23):3807–3814, 2015.

[22] Frank Yates. *The Design and Analysis of Factorial Experiments*. Imperial Bureau of Soil Science, Harpenden, England, 1937.