[Reviews · NeurIPS 2020]

Review 1

Summary and Contributions: The paper is devoted to the problem of Bayesian learning of Gaussian Bayesian networks. The authors propose several improvements with-in the MCMC framework for learning the graph in order to apply the algorithm to the bigger system. They reduce the cost of an algorithm by removing unnecessary steps, by smarter computing some quantities or by replacing full maximization by some heurestics.

Strengths: The strongest part is that a significantly reduce the cost of full Bayesian inference for DAGs.

Weaknesses: The novelty of the paper is very limited. The ais authors concentrate on computational tricks, tries to improve the scalability of the algorithm. And they achieve some success. However, for NIPS paper I would expect not only to improve implementation of the algorithm but also some new concepts. I do not found any new ideas in that sense. In more details. The novelty in GADGET algorithm are: - in preprocessing step the authors introduce reduction of the space by defining the maximum size of possible parents sets, and more sophisticated way to storage and precompute partial scores which are required further. - in MCMC step the authors modify Kuipers et al. algorithm by dividing into two steps, the first step: run MCMC only to get ordered partitions and postpone the sampling DAG step to post-processing. - in post-processing, DAG is sampled in a classical way first by adding compelled edges and next sample rest according to posterior probabilities. The algorithm 2 is a straightforward consequence of algorithm 1 except a new formula for casual effects.

Correctness: All results seem to be correct.

Clarity: I do not have major objection about the clarity of the paper.

Relation to Prior Work: The existing results and methods are well described and the authors clearly explain their contribution.

Reproducibility: Yes

Additional Feedback: #UPDATE# After reading other reviews and rebuttal. I can agree that the improvement of algorithms in terms of the computational cost is important. Due to that, I decided to increase my score.


Review 2

Summary and Contributions: The authors give optimizations for a certain kind of MEC learning of discrete models.--specifically methods that score DAGs, looking for the most optimal score, over a sequence of DAGs generated by a Markov chain, where the Markov chain is achieved in various ways. The main way to optimize this procedure, as I read the article, is to improve the Markov chain procedure itself--for example, by limiting the potential parents of a node to a set of K such potentials. There are other optimizations as well--for instance, tricks to save memory. The contribution of the work is increase the number of nodes of a problem that can be analyzed this way.

Strengths: The main strength of the work is in making explicit some of the tricks that one might use to optimize the Markov chains of DAGs considered by this kind of procedure. This will be relevant to other people trying to use these methods who are facing similar problems. Two particular heuristics are identified for larger numbers of nodes, GREEDY and BACK & FORTH, which do particularly well.

Weaknesses: Although it is hinted at in the experimental section, it is not explicitly stated that methods exist (possibly not performing as well) to infer an MEC without using the above Markov chain procedure for stopping through DAGs. Two are mentioned, PC and GES, where these use independence tests or scores over discrete variables. Whatever their accuracy, both of these methods are highly scalable, far more scalable than the TABU-style searches discussed here. To be fair, this should be stated and the explicit reason for discussing TABU-style searches indicated, since these are in fact, at least theoretically, ways of achieving scalable MEC-style analyses for discrete datasets.

Correctness: The empirical methodology is excellent. This of course is mostly in the supplement, which is very clear. The claims all appear to be correct.

Clarity: Other than the framing of it, it's fine

Relation to Prior Work: See above.

Reproducibility: Yes

Additional Feedback: The author feedback I thought was good. I'm very supportive of the idea that we need to improve performance of this particular algorithm, or class of algorithms, since I find also that it has performance advantages for accuracy.


Review 3

Summary and Contributions: This paper considers Bayesian inference for causal graphs with observational data. The first contribution is on speeding up a DAG posterior MCMC sampler. It builds upon a recent work (Kuipers et al. 2018) and proposes algorithmic improvements to reduce space and time complexity of the algorithm. Then it discusses the best candidate parent set selection problem that helps posterior coverage in a limited budget. An exact but expensive algorithm is provided, hence a number of heuristic approximations are listed and compared. Finally, an alternative to IDA is proposed for Bayesian estimation of causal effects in linear Gaussian model. Overall I feel the paper is a solid collection of results. ====== Thanks for providing additional explanations. I think computational issue is important in Bayesian method for BNSL, so I'll keep my current score. I'm looking forward to the promised updates in the final version.

Strengths: Algorithmic improvement in multiple components of the Bayesian inference pipeline.

Weaknesses: Contribution in each individual part is relatively insignificant.

Correctness: Appears to be but did not check thoroughly.

Clarity: This paper is well written.

Relation to Prior Work: Yes.

Reproducibility: Yes

Additional Feedback: - (Kuipers et al. 2018) allows one to have an extra parent outside the candidate parent set. It would be nice to have a discussion on this in the paper. - Since this is an MCMC paper, a mixing plot would be nice to show how fast the chain converges. - What value of K did different methods use? - How much benefit does large K bring in estimating causal effect? - The paper claims "can handle hundreds of variables" but the largest experiment is with 104 variables. Maybe this can be technically counted as "hundreds" but I would be more conservative in the description here. - Line 84: B is not defined, but from the context it b_{ij} is the edge weight for x_j -> x_i. - Line 91: Node 1 should be the ancestor of node 6, not the other way around. - Line 274: Incomplete sentence.


Review 4

Summary and Contributions: Like it says in the abstract there are 3 main contributions: 1. Algorithmic tricks for faster MCMC for DAGs 2. Methods for choosing a good set of potential parents for given DAG vertices 3. Application of sampling for a Bayesian measure of causal effect estimation. Experimental results supporting the effectiveness of the presented methods is given.

Strengths: Causal DAG learning remains of interest to the community. This work is more ambitious than much in this area since a Bayesian approach is taken where we must consider an entire posterior distribution over DAGs. The paper is actually *useful* in that it details key algorithmic innovations to get a method working. Useful empirical work is presented to help decide between competing reasonable choices e.g. for choosing 'good' parents.

Weaknesses: The selection of candidate parents methods were heuristic rather than based on any solid theory. But they were explicitly labelled as heuristic and experiment was used to choose between them so this is a small weakness.

Correctness: I found no technical errors and appropriate experiments were conducted.

Clarity: The writing is good and I only found one typo p142: precompue -> precompute

Relation to Prior Work: The paper explicitly builds on recent prior work and makes appropriate connections to related work. The authors also make use of existing software.

Reproducibility: Yes

Additional Feedback: As should be clear from the comments given above, I think this is a useful paper that merits acceptance. A worthwhile and difficult problem is attempted and a new and successful method is created to deal with it. I would have liked more about computing the "inverse proposal probability" in Metropolis-Hastings. Perhaps this is detailed in the associated arxiv paper, but I think it needs to be mentioned here. Also why choose randomly (I assume this means with equal probability) between moves? Is this for easy computation or because there is no particular reason to bias towards, say, splitting. Robert Cowell had some software called BAIES, see Cowell 1992 in "Bayesian statistics 4", so we have a name clash unfortunately. AFTER DISCUSSION/AUTHOR FEEDBACK Thanks to the authors for their feedback. I remain in favour of acceptance.

[Author Response · NeurIPS 2020]

1   We thank all reviewers for thoughtful feedback! We reply separately to each reviewer.

2   **Reviewer #1:** We would like to point out some of the paper's main contributions, not fully recognized in the review.

3
4     • We would like to correct that the paper is *not* devoted to Gaussian BNs: two out of the three main contributions concern any BN models with efficient-to-compute local scores (cf. Questions Q1 and Q2, Sections 3 and 4).

5
6     • We actually claim several contributions with both conceptual and technical novelty; we will better highlight these in the next version of the paper.

7
8       – One example is the definition of the maximum coverage problem (two variants), along with the associated (exponential-time) algorithm to solve the problem optimally.

9
10
11
12
13       – Another example is our algorithm for sampling DAGs conditionally on a root-partition (Sections 3.4 and A.3). Not only do we introduce a novel and somewhat generic idea of reusing space by sampling component-wise (parents for fixed node) as opposed to object-wise (whole DAGs), but we also present a technique for sampling weighted subsets of a ground set using inclusion–exclusion. As far as we know, these ideas are novel and may well have applications also beyond the context of DAG sampling.

14
15     Since the established full Bayesian framework gives us a principled machine learning approach, the essential challenges concern the computational tasks. Accordingly, our main innovations are algorithmic.

16
17     • We would like to correct that our algorithm for sampling DAGs is *not* "classical" (cf. previous item). There generally are no compelled edges, just the constraint of selecting at least one parent from the previous part.

18   **Reviewer #2:** We would like to clarify possible minor misunderstandings and further justify our chosen approach.

19     • Our approach can handle discrete and continuous data sets, and we use both extensively in our experiments.

20
21
22
23
24     • We would like to stress that we are not seeking a single optimal DAG or even MEC. Instead, we take a full Bayesian approach and perform inference based on the posterior distribution over all DAGs (in practice, a sample from the posterior). By averaging over DAGs (belonging to different MECs), we can properly account for the uncertainty related to the graph structure in any subsequent inference tasks. This is not the case when using non-Bayesian structure learning methods, such as PC and GES, which return a single DAG or MEC.

25
26
27     • Furthermore, in practice, the Bayesian approach has been shown to outperform non-Bayesian methods in accuracy in causal inference tasks [29, 1, 19]. We confirm this finding in our experiments on estimating linear causal effects (Figure 3(b–d)), comparing against both PC- and GES-based methods.

28
29
30     • Thus the explicit reasons for our aim here are methodological and related to superior accuracy performance. We will better motivate our choice of approach and highlight its advantages in contrast to PC and GES in the next version of the paper. Thank you for pointing out this need.

31   **Reviewer #3:** Thank you for your insightful comments and questions (suggesting grounds for higher confidence)!

32
33     • In Section A.4 (Supplement), we discuss how allowing parents outside the candidate set is implemented in our method, comparing it to the approach of Kuipers et al. [17].

34     • We will add mixing plots illustrating the convergence of the chains to the next version.

35
36
37
38     • The number $K$ is only relevant to Gadget and BiDAG. Both methods determine an appropriate value of $K$ based on the data set and computation time allowed. The improved accuracy of Gadget in comparison to BiDAG in Figure 3(d) is, in part, due to Gadget being able to use a larger $K$, namely $K = 15$. We will examine this observation further in the next version.

39
40     • Our intention was not to explicitly claim our method can handle hundreds of variables, although the theory (bounds in Table 1 linear in $n$) and simulations suggest so. We will change the wording in Concluding remarks.

41     • Thanks for pointing out the typos. These will be fixed in the next version of the paper.

42   **Reviewer #4:** Thank you for appreciating our ambitious Bayesian approach.

43
44     • The moves and the associated proposal probabilities are indeed described in detail in the papers presenting partition MCMC [16] and BiDAG [17].

45
46
47     • We did not analyze how much the performance of the sampler could be improved by tuning the selection probabilities among the moves, as our choices (equal probabilities) provided good performance already. This question warrants more careful analysis in future work.

48     • Thanks for the pointer on the naming conflict! We will find a new name for the next version of the paper.

[Meta-Review · NeurIPS 2020]

This paper presents a collection of useful tricks to speed up Bayesian computations for causal discovery algorithms. Despite some concerns regarding novelty, all reviewers agreed that this paper is well-written and could help spur interest and further developments in Bayesian algorithms for BNSL.